# Knowledge and stigma of autism spectrum disorders in Chinese university students in the context of inclusive education

Jinping Hu[1,2], Pengwei Fu[1]*, Shiying Qiao[2], Xinai Yan[2]

1 School of Philosophy and Social Development, Shandong University, Jinan, China, 2 School of Education and Psychology, University of Jinan, Jinan, China

* fupengwei2023@mail.sdu.edu.cn

## Abstract

This study investigated knowledge of autism spectrum disorder and associated stigma among Chinese university students, utilizing the cross-culturally validated Chinese version of the Autism Stigma and Knowledge Questionnaire. A total of 2,081 students from 25 provinces participated in an online survey. Independent-samples *t*-tests and one-way ANOVAs revealed that female students, upper-grade students, normal-education students, special education majors, and those who had completed inclusive education courses demonstrated significantly higher levels of ASD knowledge and lower levels of stigma. Prior interactions with autistic people were also associated with greater understanding of ASD and more accepting attitudes. These findings emphasize the significance of incorporating autism-related content into both general and special education curricula, fostering high-quality interactions with autistic communities, and critically considering the types of knowledge that most effectively mitigate stigma. As inclusive education reforms progress in China, enhancing professional training and awareness of neurodiversity will be crucial for preparing future educators to establish inclusive and supportive learning environments.

## Introduction

The rising prevalence of autism has become a global educational challenge, prompting an urgent need for inclusive educational strategies [1–3]. Autism Spectrum Disorder (ASD) is a neurodevelopmental condition that typically emerges in early childhood. According to a 2023 report from the U.S. Centers for Disease Control and Prevention (CDC), approximately 1 in 36 children in the United States is diagnosed with ASD [4]. In China, it is estimated that over two million children are on the autism spectrum, with approximately 150,000 new cases reported annually [5]. As more autistic children enter mainstream classrooms, inclusive education has become a

**Data availability statement:** All relevant data are included in the Supporting Information file (Data.csv). This file contains anonymized participant-level data including all variables used in the reported analyses, such as ASD knowledge and stigma scores, demographic characteristics, and computed scores for group comparisons, correlation, and regression analyses.

**Funding:** The author(s) received no specific funding for this work.

**Competing interests:** The authors have declared that no competing interests exist.

vital pathway toward equitable learning environments and improved social integration [6,7].

China has actively promoted inclusive education through policy reforms, including the 14th Five-Year Plan for Special Education Development and the Enhancement Action Plan. However, the implementation of these policies is inconsistent, and public awareness, particularly among educators, is still evolving. General education teachers are increasingly expected to not only deliver subject-specific instruction but also to support students with special needs. This shift necessitates an improvement in "inclusive education literacy," especially among pre-service teachers [8,9]. In 2021, the Chinese Ministry of Education updated national standards to formally integrate inclusive education content into teacher education programs [10].

ASD knowledge and public attitudes are crucial determinants of the success of inclusive education. Prior research has shown that insufficient understanding and the persistence of stereotypes about autism remain common among university students [11–13]. Stigma, often reflected in attitudinal negativity, social distancing, and stereotypical perceptions, remains a significant barrier to the social inclusion of autistic children [14,15]. In this study, stigma is conceptualized as attitudinal stigma, encompassing negative perceptions and emotional responses toward children with ASD. This dimension was assessed using seven binary-response items within the Autism Stigma and Knowledge Questionnaire, a validated measure [16].Higher total scores on this subscale indicate more accurate beliefs and therefore lower levels of stigma.

Some studies suggest that improving autism knowledge can reduce stigma and increase inclusive attitudes [17–19].However, misconceptions remain prevalent, with autism frequently mischaracterized as a behavioral or psychological disorder rather than understood as a form of neurodiversity. The framing of autism, whether through a pathological lens or an affirming neurodiversity perspective, may significantly influence the degree of stigma associated with the condition.

Universities play a pivotal role in preparing future educators and promoting neurodiversity-affirming values [20]. Pre-service teachers, especially those majoring in special education or who take inclusive education courses, are at the forefront of shaping inclusive classrooms. However, surveys indicate that even among education majors, there are gaps in knowledge and instances of implicit or explicit stigma [21,22].

This study aims to:

Examine ASD knowledge and stigma among Chinese university students across key demographic groups;

Assess whether education-related variables, such as academic major, prior interact with autistic people, and coursework in inclusive education, are significantly associated with higher levels of ASD knowledge and lower levels of stigma.

We hypothesize that pre-service teachers (normal education students) will demonstrate higher ASD knowledge and lower stigma compared to non-normal education students. Among them, students majoring in special education will exhibit the highest knowledge and the lowest stigma, followed by those who have completed inclusive education courses.

## Methods

### Participants

A total of 2,564 university students participated in the online survey. After excluding 483 invalid responses, which were removed due to incomplete data or failure to meet attention-check criteria, the final analytic sample consisted of 2,081 valid responses, yielding an effective response rate of 81.2%. The demographic characteristics of the participants are presented in Table 1.

To ensure data quality and minimize inattentive or random responding, three attention-check items were embedded throughout the questionnaire. These items employed direct response instructions (e.g., "Please select option 2 for this question") to verify participant engagement. An analysis of excluded cases revealed no systematic demographic bias (e.g., by gender, major, or academic grade), supporting the robustness of the retained dataset.

To explore differences in autism-related knowledge and stigma based on academic training, participants were categorized into two groups: normal-education students and non-normal-education students. In the context of Chinese higher education, normal-education students are enrolled in teacher education programs, typically at institutions referred to as "normal universities." These students are considered pre-service teachers, preparing for future careers in general education settings, such as primary and secondary schools.In contrast, non-normal-education students are enrolled in academic disciplines outside the field of education (e.g., engineering, business, law). They do not receive professional training in pedagogy or inclusive education. This classification enables a comparative analysis of how academic orientation and pre-service training may shape students' knowledge of ASD and their attitudes toward stigma.

**Table 1. Demographic statistics of participants (*N*=2,081).**

| Characteristics | | *N*(%) |
|---|---|---|
| General | Male | 563 (27.1%) |
| | Female | 1518 (72.9%) |
| Ethnicity | Han ethnic group | 2002 (96.2%) |
| | Minority | 79 (3.8%) |
| Birthplace | Urban | 759 (36.5%) |
| | Rural | 1322 (63.5%) |
| Grade | Freshman | 1145 (55.0%) |
| | Sophomore | 587 (28.2%) |
| | Junior | 246 (11.8%) |
| | Senior | 103 (4.9%) |
| Whether interact with autistic people | Yes | 320 (15.4%) |
| | No | 1761 (84.6%) |
| Whether major in special education | Yes | 209 (10.0%) |
| | No | 1872 (90.0%) |
| Whether take inclusive education courses | Yes | 148 (7.1%) |
| | No | 1933 (92.9%) |
| type of major | Normal education students | 707 (34.0%) |
| | Non Normal education Students | 1374 (66.0%) |

*Note:In the current cultural context of China, there are only two officially recognized genders. Therefore, gender is only set as male or female here;Ethnic groups are generally divided into Han and minority ethnic groups.*

## Procedures

This study was conducted following approval from the ethics review board of the first author's affiliated university. Data were collected through an online survey administered via Wenjuanxing (https://www.wjx.cn), a widely used online survey platform in China.

The survey link was disseminated through QQ and WeChat, two of the most popular social media platforms in China. QQ is an instant messaging platform developed by Tencent and is commonly used in educational settings for creating class group chats and managing academic communication. Similarly, WeChat is a multifunctional social platform frequently used for both personal and institutional communication. In Chinese higher education, both QQ and WeChat class groups serve as primary tools for student coordination, making them effective channels for participant recruitment [23].

The first author and co-authors distributed the survey link to university class groups through these platforms. Participation was entirely voluntary and anonymous. No monetary or material compensation was offered. An online informed consent form was presented at the start of the survey, clearly stating the study's purpose—to assess university students' knowledge of ASD and their attitudes toward autism-related stigma. Procedures were followed after approval from the university's review board..

## Materials

**The Autism Knowledge and Stigma Questionnaire(ASK-Q).** The questionnaire has been proven to have high internal consistency Cronbach's Alpha = 0.88 [24]. The ASK-Q was developed for cross-cultural comparison of ASD knowledge. During its development phase, international researchers evaluated the items in the ASK-Q to ensure its cross-cultural validity, reducing the need for cross-cultural adaptation in future studies [25].

The ASK-Q is a 49-item instrument designed to measure knowledge and stigma related to ASD across multiple domains. The scale includes: Diagnostic knowledge (18 items; e.g., "Some children with autism do not speak"), Causal knowledge (16 items; e.g., "Autism is a brain-based condition"), and Treatment knowledge (14 items; e.g., "Early intervention can significantly improve social and communication abilities in children with autism").Among the 48 scored items, 7 also measure attitudinal stigma toward autism, reflecting common misconceptions (e.g., "Autism is caused by cold and rejecting parents"). One general screening item ("I have heard of autism") was excluded from scoring, as it could not be evaluated as objectively correct or incorrect. Responses were scored dichotomously (1 = correct; 0 = incorrect), resulting in a total knowledge score ranging from 0 to 48, where higher scores indicate greater ASD knowledge and lower endorsement of stigmatizing beliefs. The stigma score was based on the 7 stigma-related items, with higher scores indicating lower stigma endorsement.

This study used the Chinese translated version. The Chinese version of the ASK-Q survey was meticulously translated in accordance with a rigorous translation-retranslation protocol. Dr. Y initially rendered the survey items into Chinese, followed by a retranslation by a bilingual researcher uninformed of the ASK-Q. A bilingual speech-language pathologist then assessed the linguistic accuracy of the retranslated version. Subsequently, three ASD bilingual experts with diverse backgrounds evaluated the Chinese translation against the criteria of semantic, idiomatic, experiential, and conceptual equivalence as proposed by Guillemin et al. (1993). Cultural adaptability was ascertained through unanimous approval from the reviewers. The meticulously adapted Chinese ASK-Q has demonstrated robust psychometric properties across a substantial sample of Chinese respondents [26].

**Data analysis.** The data were inputted by the researcher and subjected to cross-comparisons and logic checks prior to statistical analysis using SPSS 27.0. Initial steps included calculating means and standard deviations, followed by *t*-tests and *F*-tests to examine variable correlations. Subsequently, Pearson correlation analysis assessed the relationship between knowledge and stigma, with all tests set at a $p < 0.05$ significance level and employing two-sided tests.

To further examine the statistical independence and potential collinearity among predictor variables used in the ANOVA tests, we conducted a supplementary multiple linear regression analysis. The total score for autism knowledge and stigma were used as dependent variables in two separate models. Independent variables included gender, ethnicity, place of birth, academic grade, exposure to autistic people, whether the participant majored in special education, whether they had received inclusive education training, and whether they were enrolled in a teacher training program.

We calculated the Variance Inflation Factor (*VIF*) for each predictor to evaluate multicollinearity and applied the Durbin-Watson test to assess autocorrelation of residuals. This analysis served as a robustness check to ensure that the assumptions underlying our main statistical comparisons were not violated.

## Results

### Additional analyses for confounding effects

To account for potential confounding variables and test the robustness of the group comparisons, we conducted multiple linear regression analyses using the total ASD knowledge score and total stigma score as dependent variables. Predictor variables included gender, ethnicity, place of birth, academic grade, prior interact with autistic people, special education major status, inclusive education coursework, and enrollment in a teacher preparation program.

The regression model for ASD knowledge yielded **$R^2 = 0.186$**, indicating that 18.6% of the variance in knowledge scores could be explained by the included predictors. The model for stigma yielded **$R^2 = 0.075$**, suggesting that although smaller, a non-negligible proportion of variance was accounted for(see Tables 2 and 3).

*VIF* values ranged from 1.009 to 1.655, indicating no multicollinearity among predictors. Durbin-Watson values (1.855 for knowledge, 1.945 for stigma) showed no evidence of autocorrelation. These results confirm that key variables were statistically independent and that the ANOVA-based comparisons presented below were not confounded by multicollinearity(see Table 4).

These supplementary analyses reinforce the robustness of the group differences reported below and support the hypothesis that education-related variables independently influence ASD knowledge and stigma.

### Demographic differences in ASD knowledge and stigma

Independent-samples *t*-tests and one-way ANOVAs were conducted to examine group differences in ASD knowledge and stigma across demographic variables(see Table 5).

Gender: Female students reported significantly higher ASD knowledge ($M = 33.92$, $SD = 4.10$) than male students ($M = 31.54$, $SD = 4.27$), $t = 11.63$, $p < 0.001$, $d = 0.574$. They also endorsed significantly lower levels of stigma ($M = 3.56$, $SD = 1.54$ vs. $M = 3.10$, $SD = 1.55$), $t = 5.98$, $p < 0.001$, $d = 0.295$.

**Table 2. Model summary for regression predicting autism knowledge scores (*N* = 2,081).**

| Model | *R* | $R^2$ | Adjusted $R^2$ | Standard error of the estimate | Durbin-Watson |
|---|---|---|---|---|---|
| 1 | 0.431 | 0.186 | 0.183 | 3.87 | 1.86 |

*Note. Dependent variable: total knowledge score. R² indicates that 18.6% of the variance is explained by the model.*

*Durbin-Watson statistic indicates no autocorrelation in residuals.*

**Table 3. Model summary for regression predicting autism stigma scores (*N* = 2,081).**

| Model | *R* | $R^2$ | Adjusted $R^2$ | Standard error of the estimate | Durbin-Watson |
|---|---|---|---|---|---|
| 1 | 0.273 | 0.075 | 0.071 | 1.50 | 1.95 |

*Note. Dependent variable: total stigma score. R² indicates that 7.5% of the variance is explained by the model.*

*Durbin–Watson statistic indicates no serious autocorrelation in residuals.*

**Table 4. Regression coefficients and multicollinearity statistics for predictors of autism knowledge scores (N = 2,081).**

Coefficient[a]

| Model | Factors not normalized | | Normalization factor | t | p | Colinearity statistics | | |
|---|---|---|---|---|---|---|---|---|
| | **B** | **SE** | **β** | | **Tolerance** | **VIF** | |
| (constant) | 40.705 | 1.231 | — | 33.076 | 0.000 | — | — |
| Gender | 2.050 | 0.198 | 0.213 | 10.360 | 0.000 | 0.930 | 1.076 |
| Ethnic group | −0.373 | 0.445 | −0.017 | −0.836 | 0.403 | 0.991 | 1.009 |
| Place of birth | −0.192 | 0.177 | −0.022 | −1.082 | 0.279 | 0.985 | 1.015 |
| Grade | 0.401 | 0.118 | 0.082 | 3.411 | 0.001 | 0.687 | 1.455 |
| Whether interact with autistic people | −0.445 | 0.247 | −0.038 | −1.801 | 0.072 | 0.905 | 1.105 |
| Whether major in special education | −4.796 | 0.332 | −0.337 | −14.425 | 0.000 | 0.719 | 1.391 |
| Whether or not you have taken integrated education | −0.871 | 0.424 | −0.052 | −2.052 | 0.040 | 0.604 | 1.655 |
| Teacher professional or not | 0.401 | 0.224 | 0.044 | 1.791 | 0.073 | 0.638 | 1.567 |

[a]Dependent variable: Total score.

Note. Dependent variable: total autism knowledge score. VIF = Variance Inflation Factor; SE = standard error; β = standardized coefficient. All VIF values < 5, indicating no multicollinearity. Significant results are bolded.

Ethnicity and place of birth: No significant differences in knowledge or stigma scores were observed based on ethnicity ($t = 1.60$, $p = 0.110$) or urban vs. rural background ($t = 0.250$, $p = 0.803$ for knowledge; $t = 0.567$, $p = 0.571$ for stigma).

Interact with autistic individuals: Students with prior interaction reported significantly higher ASD knowledge ($M = 34.51$, $SD = 4.92$) than those without such experience ($M = 33.05$, $SD = 4.11$), $t = 4.99$, $p < 0.001$, $d = 0.343$, and lower stigma ($M = 3.77$, $SD = 1.71$ vs. $M = 3.37$, $SD = 1.52$), $t = 3.89$, $p < 0.001$, $d = 0.256$.

These findings support the hypothesis that demographic factors, particularly gender and prior personal experience with ASD, are significantly associated with levels of ASD knowledge and stigma among university students.

## Relationship between academic grade and ASD knowledge

A Spearman's rank-order correlation was conducted to examine the relationship between students' academic grade level and ASD knowledge. Results showed a small but statistically significant positive correlation, $\rho = 0.097$, $p < 0.01$ (two-tailed), indicating that students in higher academic years tended to report slightly greater knowledge of ASD(see Table 6).

This finding aligns with our hypothesis that academic progression contributes to increased autism-related knowledge, likely due to cumulative exposure to coursework and educational experiences.

## Education-related differences in ASD knowledge and stigma

To directly test our core hypotheses regarding educational background, a series of t-tests were conducted comparing students across key academic training variables(see Table 5):

Special education majors scored the highest in ASD knowledge ($M = 37.89$, $SD = 3.30$) and the lowest in stigma ($M = 4.56$, $SD = 1.31$), outperforming all other groups ($t = 20.81$, $p < 0.001$ for knowledge; $t = 12.87$, $p < 0.001$ for stigma), with large effect sizes ($d = 1.287$ for knowledge; $d = 0.827$ for stigma).

Inclusive education coursework: Students who had completed relevant coursework reported higher knowledge ($M = 35.26$, $SD = 4.67$ vs. $M = 33.42$, $SD = 4.17$), $t = 5.42$, $p < 0.001$, $d = 0.505$, and lower stigma ($M = 3.77$, $SD = 1.82$ vs. $M = 3.49$, $SD = 1.53$), $t = 2.34$, $p = 0.020$, $d = 0.232$.

General education majors vs non-education majors: Students in teacher preparation programs reported significantly higher ASD knowledge ($M = 34.48$, $SD = 4.40$ vs. $M = 32.65$, $SD = 4.07$), $t = 9.20$, $p < 0.001$, $d = 0.437$, and lower stigma ($M = 3.72$, $SD = 1.62$ vs. $M = 3.29$, $SD = 1.50$), $t = 5.80$, $p < 0.001$, $d = 0.275$.

**Table 5. Demographic variables and comparison of means between groups(*N* = 2,081).**

| Characteristics | | Total knowledge score (D + E + T) *M(SD)* | Stigma *M(SD)* |
|---|---|---|---|
| Gender | | $t = -11.633$ | $t = -5.979$ |
| | | $p < 0.001$ | $p < 0.001$ |
| | | $d = -0.574$ | $d = -0.295$ |
| | Male | 31.54 (4.272) | 3.1030 (1.546) |
| | Female | 33.92 (4.096) | 3.5586 (1.544) |
| Ethnicity | | $t = 1.600$ | $t = -1.076$ |
| | | $p = 0.110$ | $p = 0.282$ |
| | | $d = 0.183$ | $d = -0.123$ |
| | Han ethnic group | 33.30 (4.269) | 3.4281 (1.563) |
| | Minority | 32.52 (4.414) | 3.6203 (1.408) |
| Birthplace | | $t = 0.250$ | $t = 0.572$ |
| | | $p = 0.803$ | $p = 0.567$ |
| | | $d = 0.011$ | $d = 0.026$ |
| | Urban | 33.30 (4.284) | 3.4611 (1.589) |
| | Rural | 33.26 (4.273) | 3.4206 (1.539) |
| Grade | | $F_{(3,2077)} = 7.410$ | $F_{(3,376.176)} = 3.293$ |
| | | $p < 0.001$ | $p = 0.021$ |
| | | $\eta^2 = 0.011$ | $\eta^2 = 0.005$ |
| | Freshman | 32.93 (4.069) | 3.3520 (1.508) |
| | Sophomore | 33.47 (4.490) | 3.5315 (1.568) |
| | Junior | 33.87 (4.471) | 3.4593 (1.711) |
| | Senior | 34.50 (4.392) | 3.7573 (1.599) |
| Whether interact with autistic people | | $t = 4.989$ | $t = 3.887$ |
| | | $p < 0.001$ | $p < 0.001$ |
| | | $d = 0.343$ | $d = 0.256$ |
| | Yes | 34.51 (4.921) | 3.7719 (1.711) |
| | No | 33.05 (4.110) | 3.3742 (1.520) |
| Whether major in special education | | $t = 20.813$ | $t = 12.870$ |
| | | $p < 0.001$ | $p < 0.001$ |
| | | $d = 1.287$ | $d = 0.827$ |
| | Yes | 37.89(3.297) | 4.5598(1.307) |
| | No | 32.76(4.058) | 3.3098(1.532) |
| Whether take inclusive education courses | | $t = 5.421$ | $t = 2.344$ |
| | | $p < 0.001$ | $p = 0.020$ |
| | | $d = 0.505$ | $d = 0.232$ |
| | Yes | 35.26 (4.665) | 3.7703 (1.822) |
| | No | 33.12 (4.207) | 3.4097 (1.533) |
| type of major | | $t = 9.202$ | $t = 5.801$ |
| | | $p < 0.001$ | $p < 0.001$ |
| | | $d = 0.437$ | $d = 0.275$ |
| | Normal education students | 34.48 (4.403) | 3.7157 (1.620) |
| | Non Normal education Students | 32.65 (4.073) | 3.2911 (1.504) |

*Demographic variables and between-group mean comparison D = diagnosis, E = etiology, T = treatment, *p < 0.05,**p < 0.01.*

*(Note:In the Chinese context, the division regarding ethnicity is generally Han Chinese and ethnic minorities.The place of birth, i.e., the civil registry, is divided into urban and rural areas).*

**Table 6. Spearman's rank-order correlation between grade level and total ASD knowledge score (N = 2,081).**

| Variable | Grade level |
|---|---|
| Total Knowledge Score | 0.097** |

Note. Spearman's $\rho$ = .097. $p$ < 0.01 (two-tailed). A positive value indicates that higher academic grade levels are associated with higher autism knowledge scores.

These results strongly support our central hypothesis that education-related training,especially in special education and inclusive pedagogy,is associated with increased autism knowledge and reduced stigma.

## Sources of ASD knowledge among University students

As shown in Fig 1,the internet (65%)and TV/movies (64%) were the most common sources of ASD knowledge, followed by social media (48%), news reports (41%), personal experience (39%), university courses (36%), doctors or other medical professionals (21%), research articles (18%), and club activities (15%).

This distribution indicates that while digital media remains a dominant information source, formal university instruction still plays a meaningful role in shaping ASD knowledge.

## Discussion

### Controlling for confounding variables

While the primary analyses relied on ANOVA to examine group-level differences in ASD knowledge and stigma, we recognize that such analyses do not account for potential confounding factors. To address this limitation, we conducted supplementary regression analyses incorporating demographic and academic variables.

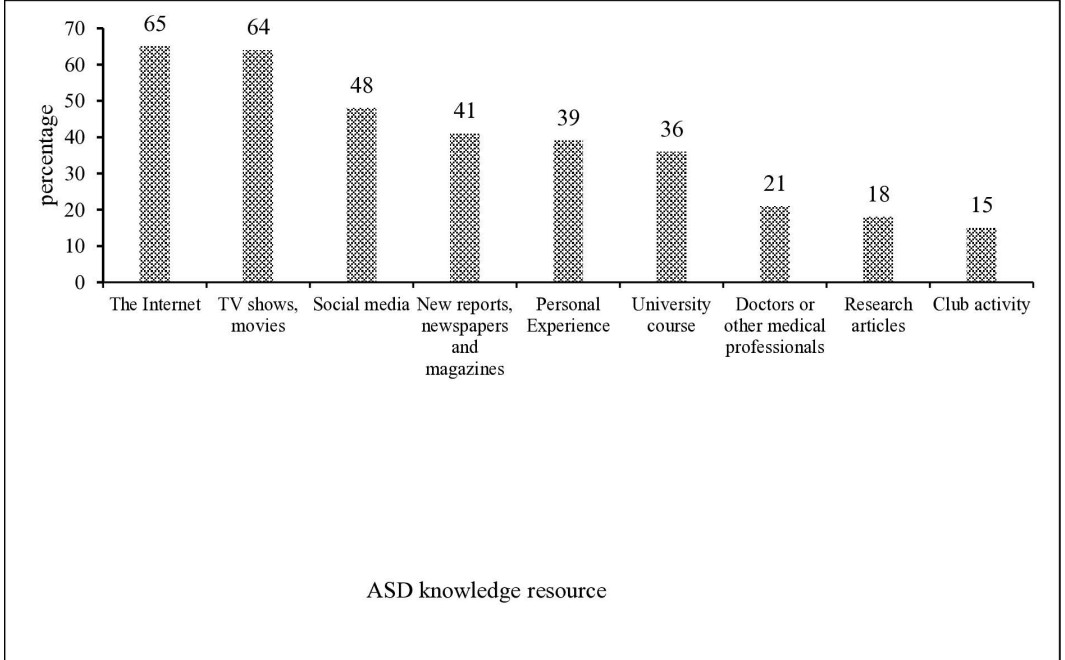

**Fig 1. Percentage of students' ways of acquiring ASD knowledge resources.**

The robustness checks confirmed that the predictors were statistically independent, with no significant multicollinearity or autocorrelation. Although the proportion of variance explained was relatively modest, particularly in the stigma model, the results nonetheless support the validity of the observed group differences.

One limitation of the present study is that participants were not asked whether they themselves identified as autistic. While autism diagnoses in China remain largely focused on children and formal diagnostic pathways for adults are limited, the presence of undiagnosed or late-diagnosed autistic students cannot be ruled out. Future research should consider incorporating diagnostic status or self-identification to better include neurodivergent voices in university settings.

Another limitation concerns potential self-selection bias. Although WeChat and QQ are widely used across Chinese universities and internet access is nearly universal among students, it is possible that individuals with less interest in autism-related topics were less inclined to participate in the survey. This may have led to a sample that is somewhat more informed or invested in the subject than the general student population. Future studies should consider adopting more diverse recruitment strategies, such as random sampling within course rosters or stratified outreach across various academic departments, to more comprehensively capture the full spectrum of student perspectives on ASD.

### Demographic influences on ASD knowledge and stigma

Consistent with prior research [17,26,27], female students in this study reported significantly greater autism knowledge and lower stigma than male students. This may reflect broader gender-related trends in empathy, helping behavior, and attunement to psychological issues. Academic grade level was also positively associated with ASD knowledge and more favorable attitudes, suggesting that cumulative exposure to educational content and social experiences over time may foster deeper understanding and inclusivity.

Prior interpersonal contact with autistic individuals was another important predictor of both greater knowledge and reduced stigma, supporting the contact hypothesis [28]. However, the literature indicates that not all contact experiences are equally beneficial. Superficial, involuntary, or unsupported interactions may reinforce rather than reduce stigma [29,30]. These findings highlight the importance of examining the quality, depth, and context of interpersonal contact in future research.

### Educational background and professional preparation

The current results strongly support the hypothesis that structured educational training, especially in special education and inclusive pedagogy, is linked to higher ASD knowledge and lower stigma. Students majoring in special education exhibited the highest knowledge scores and the lowest stigma, with large effect sizes. This aligns with prior findings emphasizing the role of targeted disability coursework and experiential learning in shaping more informed and empathetic attitudes [24].

Similarly, students who had completed coursework in inclusive education also demonstrated more favorable outcomes, although the effect sizes were smaller. These findings suggest that even limited exposure to inclusive frameworks, such as Universal Design for Learning or collaborative instructional approaches, can meaningfully enhance awareness and promote positive shifts in attitudes. The performance of general education majors, particularly pre-service teachers, further highlights the importance of incorporating neurodevelopmental content throughout the broader spectrum of teacher education curricula.

### Nature and sources of autism knowledge

Beyond educational training, the source and type of autism knowledge itself may shape stigma-related attitudes. While our data showed that the Internet, social media, and TV/movies were the most commonly cited sources of ASD knowledge, these platforms vary widely in credibility and content. Although easily accessible, such sources may inadvertently reinforce stereotypes or oversimplified views of autism [31,32].

Importantly, autism knowledge is not a uniform construct, as it may originate from clinical information such as diagnostic criteria, formal educational instruction, or lived experience narratives. These sources can convey either conflicting or complementary messages, and their influence on stigma may vary accordingly. For example, knowledge framed within deficit-based medical models often emphasizes impairments, while neurodiversity-affirming perspectives focus on individual strengths, variability, and personhood. This conceptual distinction warrants further empirical investigation. Accordingly, we recommend that future research examine which types and sources of autism knowledge are most effective in reducing stigma, as well as how students interpret and engage with these differing perspectives.

## Implications for knowledge dissemination and practice

Our findings suggest several practical pathways for improving autism understanding and reducing stigma in higher education. While informal sources such as the Internet and social media dominate, more emphasis should be placed on formal, evidence-based instruction delivered through university curricula. Additionally, the underrepresentation of medical professionals and academic research as cited knowledge sources suggests an opportunity to strengthen ties between students and professional expertise.

We recommend enhancing autism-related content within both general and specialized teacher education programs, promoting student engagement in autism-focused research and community outreach, and incorporating diverse expert perspectives into coursework, including those of autistic self-advocates, clinical psychologists, and specialists in inclusive education. Such efforts are expected to facilitate the development of both accurate knowledge and critical reflection, enabling students to understand autism through inclusive and human-centered frameworks.

## Conclusion

This study contributes to the growing literature on autism knowledge and stigma by offering empirical evidence from a large sample of Chinese university students in the context of inclusive education. The findings confirm that key demographic variables, including gender, academic year, and prior contact with autistic individuals, are significantly associated with students' knowledge of ASD and their attitudes toward stigma.

More importantly, the study provides robust support for the hypothesis that educational background and training, particularly in special education and inclusive pedagogy play a decisive role in shaping students' understanding and acceptance of autism. Special education majors exhibited the highest levels of knowledge and the lowest levels of stigma, followed by students who had completed coursework in inclusive education. These results underscore the importance of embedding ASD related content in teacher preparation curricula, not only within special education but also in general education programs.

The study also highlights the critical role of information sources in shaping autism awareness. While digital media remains the dominant channel, its quality varies. This calls for more strategic efforts to integrate formal, evidence-based, and neurodiversity-affirming content into university education and public communication platforms. Furthermore, future research should go beyond simply measuring the amount of knowledge and instead examine the types, quality, and influence of various knowledge sources, such as clinical, educational, and lived experience, in reducing stigma.

Taken together, these findings offer both theoretical insights and practical recommendations for advancing inclusive education, enhancing teacher preparation, and fostering a more informed and compassionate university culture around neurodiversity in China and beyond.

## Supporting information

**S1 File. Data.**
(ZIP)

## Author contributions

**Data curation:** Jinping Hu, Pengwei Fu, Shiying Qiao, Xinai Yan.

**Formal analysis:** Shiying Qiao, Xinai Yan.

**Investigation:** Jinping Hu, Shiying Qiao, Xinai Yan.

**Methodology:** Jinping Hu.

**Project administration:** Jinping Hu.

**Software:** Jinping Hu, Shiying Qiao, Xinai Yan.

**Writing – original draft:** Jinping Hu, Pengwei Fu, Xinai Yan.

**Writing – review & editing:** Jinping Hu.

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
