## [Decision Letter · Decision Letter 0]

Dear Dr. Hu,

Thank you for submitting your manuscript to PLOS ONE. After careful consideration, we feel that it has merit but does not fully meet PLOS ONE’s publication criteria as it currently stands. Therefore, we invite you to submit a revised version of the manuscript that addresses the points raised during the review process.

We look forward to receiving your revised manuscript.

Kind regards,

Lubna Shirin, Ph.D, M.Phil, MBBS

Academic Editor

PLOS ONE

Journal Requirements:

4. Please ensure that you include a title page within your main document. You should list all authors and all affiliations as per our author instructions and clearly indicate the corresponding author.

Additional Editor Comments:

The author can revise the manuscript and resubmit the manuscript.

Reviewers' comments:

Reviewer's Responses to Questions

**Comments to the Author**

1. Is the manuscript technically sound, and do the data support the conclusions?

Reviewer #1: No

Reviewer #2: Partly

2. Has the statistical analysis been performed appropriately and rigorously?

Reviewer #1: No

Reviewer #2: I Don't Know

3. Have the authors made all data underlying the findings in their manuscript fully available?

Reviewer #1: Yes

Reviewer #2: No

4. Is the manuscript presented in an intelligible fashion and written in standard English?

Reviewer #1: No

Reviewer #2: Yes

Reviewer #1: This study explores the knowledge and stigma of autism spectrum disorders (ASD) among Chinese university students, an important and timely topic in the field of inclusive education. The research is based on a large dataset (N=2,081). However, several methodological, statistical, and presentation issues deemed the manuscript not to be considered for publication in PLOS ONE.

1. The manuscript is not statistically sound. ANOVA is used to compare groups, but it does not control for confounding variables like gender, major, or prior ASD interaction. Regression models should be applied to ensure robustness.

2. The authors did not mention any selection criteria and sampling method. Again the manuscript states that data was collected via WeChat/QQ social groups, but this method may exclude students with limited internet access or those less interested in ASD topics, leading to self-selection bias.

3.The manuscript claims to use a Chinese version of the ASK-Q, but no details are provided on validation. How was it translated and tested for reliability? If it was previously validated, a citation should be provided.

4.The introduction discusses stigma but does not define how it was conceptualized and measured. Was stigma measured attitudinally, behaviorally, or structurally?

5. The manuscript claims a "positive correlation" between grade level and ASD knowledge scores (p<0.001). However, ANOVA does not measure correlation—a Pearson or Spearman correlation test should be conducted instead.

6. The manuscript only reports p-values but does not provide effect sizes (Cohen’s d, η², or R²). Effect sizes should be reported to show practical significance.

Reviewer #2: Thank you for the opportunity to review this manuscript, which investigates knowledge and stigma surrounding autism among Chinese student populations. I found the paper to be interesting and important, and I believe it has strong potential for publication following minor revisions. One of the strengths of the paper is its succinctness; I therefore offer my suggestions with the intention of supporting clarity, conceptual precision, and consistency—both in terms of scientific framing and language.

I have divided my comments into two broad sections: (1) Scientific and Conceptual Feedback and (2) Spelling and Grammar.

Please note that I have not been able to locate the dataset associated with this manuscript. As such, I have responded “no” to the data availability question in the review form. If the dataset is available for review or will be made available upon publication, this should be clearly stated in the manuscript and/or accompanying materials.

1. Scientific and Conceptual Feedback

A. Overall Framing and Language

The introduction could benefit from a more structured and purposeful opening. I suggest beginning with a clear statement of the broader issue—namely, the increasing number of autistic children requiring educational support—and then narrowing in on the implications for teacher preparedness and societal knowledge about autism. This would help establish a strong rationale for why autism knowledge and stigma in student populations is a crucial area of study, particularly in light of government initiatives to promote inclusive education.

The manuscript refers to autism as a “mental health issue.” While there is ongoing debate around the best way to frame autism, many autistic people and advocates regard it as a naturally occurring neurotype rather than a mental health condition. Framing autism as a disorder or pathology can contribute to stigma, particularly in the context of a paper explicitly addressing stigmatizing attitudes. Given the topic of the study, it is important to critically reflect on the terminology used, including the continued reference to “Autism Spectrum Disorder,” and consider aligning with neurodiversity-affirming language where appropriate.

There’s a sentence about the CDC prevalence data—please clarify what country this data refers to, especially since the study is based in China. It might also be worth including a sentence or two earlier on to set the Chinese educational context around autism (e.g., whether inclusive education is standardised, common, or still developing).

B. Clarity in Methods and Measures

The classification of participant groups, beginning around line 52, requires significant clarification. The terms “normal-education students” and “non-normal-education students” are unclear and potentially problematic. It is essential to specify whether these refer to:

• Student teachers preparing to teach in general versus special education settings;

• Students with versus without special educational needs;

• Or another classification altogether.

Furthermore, the term “subjects” should be replaced with “participants,” not only because it is a more respectful and contemporary term, but also to avoid confusion with “subjects” in the academic sense (e.g., disciplines or courses of study).

It wasn’t clear to me whether participants were asked if they themselves are autistic—if not, it might be worth noting this as a limitation.

The phrase “QQ social group” on line 83 might not be familiar to international readers—could you clarify what this means? (e.g., a social media platform?)

The ASK-Q scale should be cited on first mention.

It would also be helpful to know whether participants were compensated in any way.

The methods section is currently a bit dense—perhaps consider splitting it into subsections like Participants, Materials, Procedure, and Data Analysis to make it easier to follow.

C. Results, Analysis, and Interpretation

Some of the p-values were hard to interpret (e.g., lines 112–114)—I wasn’t always sure what the test was comparing (e.g., is the p-value comparing stigma levels between groups, or knowledge levels?). It might be helpful to make these tests clearer in the text.

In terms of formatting, be consistent with the number of decimal places, and spell out all statistics.

The number of participants who failed attention checks seems relatively high in proportion to the total sample. This section would benefit from context—e.g., how many checks were included, what they involved, and whether there were any identifiable patterns in the failures (e.g., demographic skew). This would help reassure readers of the robustness of the remaining dataset.

In the results section, some of the analyses and findings could be linked back more clearly to the hypotheses or research questions.

D. Discussion and Implications

The discussion section has some nice points, especially around the importance of exposure and experience in reducing stigma. However, I’d be a bit cautious with how some findings are phrased.

It might also be worth commenting on how autism knowledge is often contested or variable—some sources of “knowledge” (e.g., clinical versus lived experience accounts) may convey very different messages. You could briefly acknowledge this and suggest future research could look more closely at what kind of knowledge reduces stigma most effectively.

2. Spelling and Grammar

While the manuscript is generally clearly written, there are a few minor spelling and formatting issues that could be addressed:

• Line 24: “education’s” – missing the n.

• Lines 100–101: Missing space between words.

• Line 113: “the” may be unnecessary or redundant in this context.

• Line 204: “Reference” should be corrected to “References.”

• Table 1: “female” missing an e.

• Table Headings: Ensure consistent capitalisation across all tables.

**Do you want your identity to be public for this peer review?** For information about this choice, including consent withdrawal, please see our Privacy Policy

Reviewer #1: **Yes: ** M Tanveer Hossain Parash

Reviewer #2: No

---

## [Author Response · Author response to Decision Letter 1]

31 May 2025

Dear Editor and Reviewers,

Thank you very much for your time and constructive feedback. We have carefully addressed all the comments provided by the reviewers and the editor. Substantial revisions have been made to improve the clarity, academic rigor, and overall quality of the manuscript.

We have uploaded a detailed “Response to Reviewers” document, in which we respond point-by-point to each comment and indicate how the manuscript was revised accordingly. All changes made in the revised manuscript are clearly marked using the “Track Changes” function (see the file titled “Revised Manuscript with Track Changes”).

We sincerely appreciate the opportunity to revise our manuscript and look forward to your further review.

Kind regards,

Jinping Hu

---

## [Editor Report · Decision Letter 1]

Knowledge and stigma of autism spectrum disorders in Chinese university students in the context of inclusive education

PONE-D-25-07208R1

Dear Dr. Jinping Hu, 

We’re pleased to inform you that your manuscript has been judged scientifically suitable for publication and will be formally accepted for publication once it meets all outstanding technical requirements.

Kind regards,

Lubna Shirin, Ph.D, M.Phil, MBBS

Academic Editor

PLOS ONE
---

## [Editor Report · Acceptance letter]

PONE-D-25-07208R1

PLOS ONE

Dear Dr. Hu,

I'm pleased to inform you that your manuscript has been deemed suitable for publication in PLOS ONE. Congratulations! Your manuscript is now being handed over to our production team.

Kind regards,

on behalf of

Dr. Lubna Shirin

Academic Editor

PLOS ONE